# Misleading graphs in context: Less misleading than expected

**Jannetje E. P. Driessen**[1], **Daniël A. C. Vos**[2], **Ionica Smeets** [1], **Casper J. Albers** [2]*

**1** Science Communication and Society, Leiden University, Leiden, Netherlands, **2** Department of Psychology, University of Groningen, Groningen, Netherlands

* c.j.albers@rug.nl

**Data Availability Statement:** The data have been uploaded as part of the submission. Upon acceptance, we will put the data in a public repository or upload it as supplemental material to the paper; whatever the editor prefers.

## Abstract

Misleading graphs are a source of misinformation that worry many experts. Especially people with a low graph literacy are thought to be persuaded by graphs that misrepresent the underlying data. But we know little about how people interpret misleading graphs and how these graphs influence their opinions. In this study we focus on the effect of truncating the y-axis for a line chart which exaggerates an upgoing trend. In a randomized controlled trial, we showed participants either a normal or a misleading chart, and we did so in two different contexts. After they had seen the graphs, we asked participants their opinion on the trend and to give an estimation of the increase. Finally we measured their graph literacy. Our results show that context is the only significant factor in opinion-forming; the misleading graph and graph literacy had no effect. None of these factors had a significant impact on estimations for the increase. These results show that people might be less susceptible to misleading graphs than we thought and that context has more impact than a misleading y-axis.

## Introduction

Graphs can misrepresent the underlying data in many ways, and this worries experts in different fields, from health communication [1] to data visualization [2].

Misleading graphs were omnipresent in the media during the covid pandemic, both in (social) media and governmental communications [3, 4]. Earlier it was shown that even in the scientific journal 'Science' 30% of all graphs contained errors [5], and a later study showed that 31% of the graphs in the Journal of American Medicine were ambiguous [6].

There are many ways for a graph to be 'wrong'. For example, there can be inconsistencies within a graph, such as a $y$-axis label contradicting the title or the $y$-axis being flipped [7]. There have been many attempts to categorize misleading graphs. For instance, [8] considers seven types of distortion where the first type is manipulating scale ratios: for instance, by truncating a y-axis such that it does not start at zero. A recent overview [2] also includes this error, along with many others. This study will focus on one specific form of misleading graphs: line charts that truncate the y-axis. We focus on line charts as these are some of the most commonly used types of visualization [9].

**Funding:** The author(s) received no specific funding for this work.

**Competing interests:** The authors have declared that no competing interests exist.

## Graph literacy

Graph literacy is the individual's ability to read, process, and comprehend data visualisations [10]. In general, people interpret graphs in a few steps [11]. At first, they identify the visual features, such as the direction or colour of a line. Secondly, they interpret the relations that the visual features represent. Lastly, they match these interpretations with the labelled variables.

Individuals with higher graph literacy can process graphs with greater ease and have an expected higher comprehension of the graph's content [10]. Eye-tracking studies confirmed this effect and moreover found that people with low graph literacy have an overreliance on spatial-to-conceptual mappings, whilst people with high graph literacy spend more time looking at features such as the numbers on the scales and the axis labels [7].

There are multiple tests available to determine the level of graph literacy of an individual. For example, the objective graph literacy scale [12] is a test with 13 items. It is a reliable indicator of an individual's graph literacy level, but it takes around ten minutes to complete, which is not manageable for some studies. The four-item short graph literacy test scale (SGL) [13] is a shortened version of the objective graph literacy scale with still sufficiently good psychometric properties.

## The interpretation of misleading graphs

A few studies have studied the interpretation of conflicted graphs or other types of misleading graphs, which usually covered only bar charts. For instance, [14] researched bar graphs' interpretation when shifting the $y$-axis, but with a sample size of only nine participants. [15] studied tilted bar charts and found that these seem to be interpreted similarly to the original bar charts. An older study that used different types of misleading bar charts showed that these improperly designed charts can influence decision making [16]. [10, 17] studied the interpretation of graphs with 3D effects and concluded that in general 2D bar charts result in higher comprehension than 3D bar charts.

Experiments showed that individuals with high graph literacy have a more accurate interpretation of self-conflicting graphs [7]. [18] found that bar charts with a truncated y-axis were found to be significantly less credible than proper graphs, but the information reproduction of skewed graphs was significantly higher.

## Graphs in context

In real-life situations, graphs are used in a context, and we know that prior knowledge about a subject influences the reader's interpretation of graphs [19]. The study by [18] also looked at graphs in different contexts: both deceptive and non-biased graphs were shown in a general news and a political news setting to communication students. In both settings, they found that 'a student sample was effectively unfazed by a deceptive graphic'. They recommend that their study should be replicated in a random, representative sample.

It is known that people who do not have a strong opinion on a subject are more susceptible to persuasion using charts [20]. Therefore we decided to do a follow-up study to [18] with a random, representative sample and graphs in a made-up context that people would not have existing strong opinions on. Furthermore, we focus on line graphs since they are understudied in the current literature.

## Research question and hypothesis

In this paper, we study the interpretation of line graphs. Specifically, we study whether and how truncating the $y$-axis and the context the graph is presented in influence the

interpretation. We check whether this influence is moderated by graph literacy. Our expectation is that (i) non-shifted line graphs are better interpreted than shifted line graphs, (ii) that the context influences the judgement, and (iii) that these relations are stronger in participants with low graph literacy.

## Methods

### Design

The current study uses a quantitative survey-based research design with four distinct surveys. The surveys can be separated on two between-person factors, namely, context and graphical representation.

The contextualisation presents the participants a narrative that focuses on the fictional 'Bluebeak', a non-native bird in Denmark. One story about the 'Bluebeak' is that the bird is endangered, whereas the other story focuses on the disruption the 'Bluebeak' causes to Denmark's eco-system. Both narratives are provided in S1 Appendix.

The graphical design element accompanies the 'Bluebeak' contextualisation, where one group of each contextualisation gets a different graph presented. The first possible graph being a line graph at which the $y$-axis starts at zero, whilst the other group gets a line graph that starts two standard deviations below the mean (in line with the suggestion by [14]). However, apart from the scaling of the vertical axis, both graphical representations are the same (see Fig 1).

### Population & sample assignment

Data collection is done through Prolific (www.prolific.co), which uses representative samples from adult US citizens to fill out surveys and compensate them for their time. Prolific randomly assigns each participant to one of the four groups. The questionnaire (see S1 Appendix) is filled in unsupervised and online through Prolific.

Participants provided written informed consent. The Ethics Committee Psychology of the University of Groningen has approved this study (PSY-1920-S-0441).

### Measurement

**Variables.** The study uses two grouping factors, contextualisation and graphical representation. Additionally, we ask the participants to make a judgement call about the displayed situation, with scores ranging from 1 (very bad) to 5 (very good).

Participants are asked to estimate the proportional increase of the number of 'Bluebeaks' in the previously presented graph to determine which type of graph the participants give the most accurate proportional increase estimation. Graph literacy is measured using the SGL scale (see: Instruments) to determine whether varying levels of graph literacy influence their judgement and estimation of Denmark's 'Bluebeak' situation.

Finally, a set of demographics are included to have options for control and evaluations of sample balance. The demographics are age, gender, and educational level. Age is measured on a categorical scale ranging from 1 (18–25) to 7 (66 +), and is assumed to be continuous within the analysis. Gender has options male, female and other. Education level is measured on a 7-point scale.

**Instruments.** The short graph literacy scale consists of four items with different types of graphs displayed [13]. For instance, a pie chart is displayed and the reader is asked to assess the size of a slice of the pie. The possible scores on the short graph literacy scale range from 0 to 4, where zero indicates low graph literacy and a score of four indicates high graph literacy.

(A)

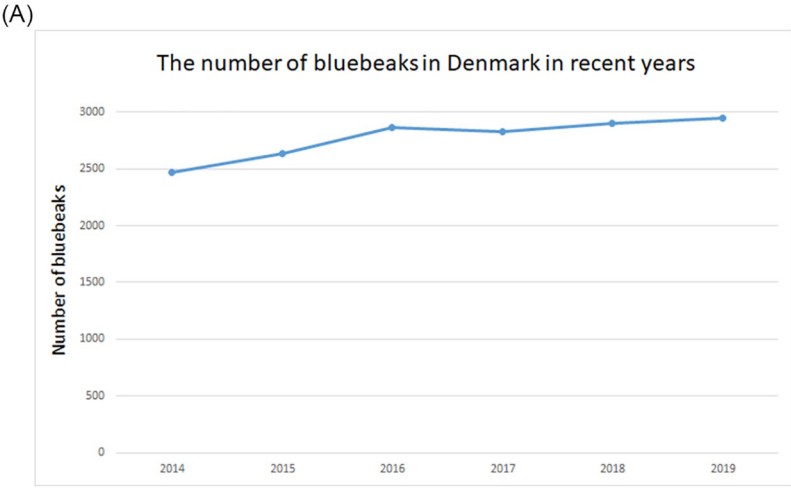

(B)

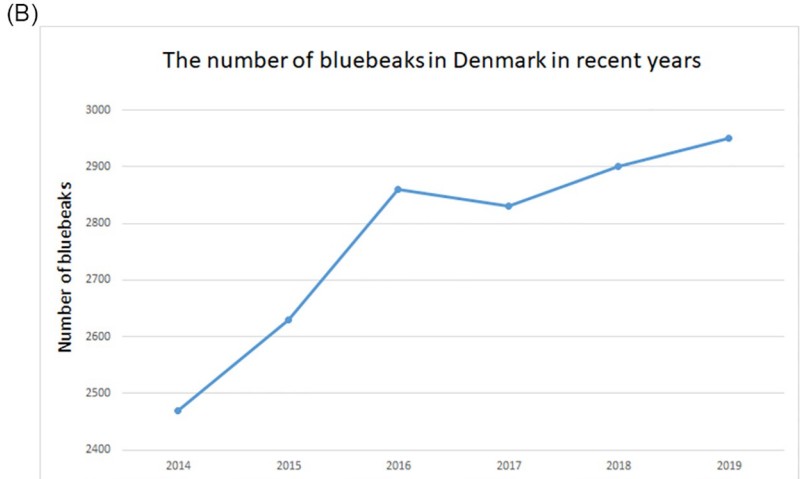

**Fig 1. The two alternative visualizations.**

**Data analysis plan.** Descriptive statistics for all variables are provided. Outliers are detected and removed based on two criteria. To avoid non-serious participants, all those with a participation duration exceeding the mean duration plus three times the standard deviation are excluded. Furthermore, outliers are detected using the MAD (median absolute deviation) on the percentage estimation variable since this is the sole variable that took manual input from the participant. For the MAD, the threshold of 2.5 is chosen, which is classified as a mildly conservative criterion [21].

The relation between axis type, context, judgement and interpretation is analysed via multiple linear regression. Two sets of models are built, one using judgment and the other using estimated proportional increase as the dependent variable. The percentage variable is centred around the correct answer; this way, the estimates received by the model are interpretable as the amount which the given estimates diverge from the correct answer.

For both sets of models, we make versions with (i) the main effects of axis-type and context, (ii) as (i) but with the inclusion of SGL as moderation variable, (iii) as (ii) but including the interaction between axis type and context. In all three models, age, education level, and gender will be included as covariates.

**Table 1. Descriptive statistics (mean score and standard deviation; the percentage for 'male') of the demographic characteristics.**

| Variable | Normal graph | | Shifted graph | |
|---|---|---|---|---|
| | *Invasive* | *Endangered* | *Invasive* | *Endangered* |
| Age | 4.14 (1.92) | 4.41 (1.78) | 3.53 (1.84) | 3.90 (1.99) |
| Education | 3.56 (1.80) | 4.20 (2.27) | 3.43 (1.87) | 3.95 (2.21) |
| Male | 48% | 47% | 60% | 47% |
| *n* | 63 | 71 | 53 | 60 |

**Power and sample size.** A posthoc power analysis is conducted for 247 participants, at which the multiple regression with $\alpha$ = .05 can discover slopes with small effects $f^2$ = .05 with a power of .968.

## Results

### Descriptive statistics

In total, 313 people started with the questionnaire. A total of 66 participants were excluded due to the following reasons: not completing the questionnaire (13), not providing consent (1), exceeding our participation duration threshold (5), not specifying gender (6) and providing scores exceeding the MAD-threshold (41). Data for 247 participants were included. Descriptive statistics are provided in Tables 1 and 2 for these *n* = 247 participants.

None of the bivariate Pearson correlations between pairs of the variables in Table 2 is significant. (*r*(judgment, percentage) = -0.002, *t*(245) = -.0026, *p* = .979, 95% CI [-0.12; 0.123]; *r*(judgment, SGL) = 0.059, *t*(245) = 0.929, *p* = .353, 95%CI [-0.066; 0.183]; *r*(percentage, SGL) = -0.000, *t*(245) = -0.005, *p* = .996, 95%CI [-0.125; 0.124]), indicating no direct linear relations.

### Linear regression

Two multiple regression models are fitted to assess the possible association between variables, one with judgment and the other with the centred percentage estimate as the dependent variable. For both models, the underlying statistical assumptions have been checked, and there were no reasons to abandon the choice for multiple linear regression. The results are given in Tables 3 and 4.

As predictor variables, the model has the type of graph (shifted vs non-shifted), the type of context (invasive vs endangered), the score on the short graph literacy (SGL) test and its interaction with the type of graph, and the demographic variables education level, age and gender.

When looking at the model with judgment as the dependent variable, context clearly is significant, with the average score in the endangered group 1.86 points higher than the ecosystem group (95% CI [1.64, 2.07], p < .001). None of the other variables is significant. When SGL is not included in the model, graph type is significant (with shifted graphs scoring, on average,

**Table 2. Descriptive statistics of the other variables.**

| Variable | Normal graph | | Shifted graph | |
|---|---|---|---|---|
| | *Invasive* | *Endangered* | *Invasive* | *Endangered* |
| Judgement | 2.21 (0.70) | 4.07 (0.76) | 2.43 (0.95) | 4.28 (0.78) |
| Percentage | 21.52 (11.10) | 17.31 (10.31) | 29.21 (15.13) | 28.37 (13.42) |
| SGL | 1.65 (0.48) | 1.80 (0.40) | 1.72 (0.50) | 1.77 (0.46) |
| *n* | 63 | 71 | 53 | 60 |

**Table 3. Model summary for the multiple linear regression model with judgment as the dependent variable.**

| Predictors | Estimate | 95% CI | p |
|---|---|---|---|
| Graph[A] | 0.38 | [-0.43; 1.19] | .359 |
| Story[B] | 1.86 | [1.64; 2.07] | < .001 |
| Education[C] | | | |
| High school | -0.28 | [-0.98; 0.42] | .430 |
| Secondary | -0.06 | [-0.75; 0.62] | .861 |
| Technical | 0.06 | [-0.67; 0.79] | .872 |
| Undergraduate | -0.30 | [-1.20; 0.59] | .504 |
| Graduate | -0.37 | [-1.52; 0.77] | .523 |
| Doctorate | 0.01 | [-0.71; 0.73] | .974 |
| Age[D] | | | |
| 26–30 | 0.13 | [-0.27; 0.53] | .527 |
| 31–35 | 0.04 | [-0.37; 0.45] | .836 |
| 36–45 | -0.01 | [-0.40; 0.38] | .948 |
| 46–55 | 0.03 | [-0.34; 0.41] | .857 |
| 56–65 | 0.05 | [-0.37; 0.46] | .820 |
| 66+ | 0.05 | [-0.37; 0.47] | .825 |
| Gender | 0.05 | [-0.16; 0.26] | .643 |
| SGL | -0.07 | [-0.39; 0.26] | .690 |
| Graph*SGL | -0.07 | [-0.53; 0.38] | .745 |
| $R^2$ | .596 | | |
| Adj. $R^2$ | .566 | | |
| n | 247 | | |

[A] normal graph is the reference group

[B] Invasive is the reference group

[C] No formal education is the reference group

[D] 18–25 is the reference group

0.25 higher than normal graphs (95% CI [.03, .46], p = .023), but this effect vanishes when graph literacy is included.

A similar picture arises when looking at the model with the percentage estimate centred around the correct answer (approximately 16%) as the dependent variable. Here none of the variables is significant. (The age group 46–55 differs (p = .039) from the reference group, but this effect is non-significant when correcting for multiple comparisons.) When SGL is excluded from the model, the variable graph does become significant, with shifted graphs scoring 9.12 percentage points higher than non-shifted graphs (95% CI [5.86, 12.39], p < .001) but also here, the effect vanishes when SGL is included in the model.

## Conclusion & discussion

This study looks at how a truncated y-axis, context, graph literacy and demographics influence the judgment and interpretation of graphs.

When participants are required to make a judgement call based on a displayed graph, it becomes apparent that the story itself matters more in the decision-making process than the shape of the graph's *y*-axis. This is in contrast to the conclusions of [22], who reported that distorted graphs influence both judgement and comprehension. The deviating results could be due to different choices in the operationalisation of the studies. For instance, [22] studied bar graphs, whereas we focused on line graphs.

**Table 4. Model summary for the multiple linear regression model with percentage as the dependent variable.**

| Predictors | Estimate | 95% CI | p |
|---|---|---|---|
| Graph[A] | 5.68 | [-6.86; 18.22] | .373 |
| Story[B] | -2.78 | [-6.04; 0.48] | .095 |
| Education[C] | | | |
| High school | 1.14 | [-9.73; 12.00] | .837 |
| Secondary | -3.05 | [-13.64; 7.54] | .571 |
| Technical | -1.94 | [-13.20; 9.32] | .734 |
| Undergraduate | -0.90 | [-14.71; 12.91] | .898 |
| Graduate | -10.83 | [-28.56; 6.90] | .230 |
| Doctorate | 0.64 | [-10.47; 11.74] | .910 |
| Age[D] | | | |
| 26–30 | -1.09 | [-7.31; 5.14] | .731 |
| 31–35 | -4.84 | [-11.16; 1.49] | .133 |
| 36–45 | -4.11 | [-10.16; 1.93] | .181 |
| 46–55 | -6.09 | [-11.86; -0.32] | **.039** |
| 56–65 | -5.47 | [-11.89; 0.96] | .095 |
| 66+ | -2.29 | [-8.77; 4.19] | .487 |
| Gender | -0.04 | [-3.25. 3.17] | .979 |
| SGL | 0.46 | [-4.54; 5.46] | .857 |
| Graph*SGL | 1.96 | [-5.03; 8.91] | .582 |
| $R^2$ | .184 | | |
| Adjusted. $R^2$ | .123 | | |
| n | 247 | | |

[A] normal graph is the reference group

[B] Invasive is the reference group

[C] No formal education is the reference group

[D] 18–25 is the reference group

For the interpretation of the graph, truncating the y-axis does have a significant impact on percentage estimates, which is in line with [14], who claimed that the y-axis of a line graph should start at least 1.5 standard deviations to be interpreted correctly. However, we see that the influence of the truncated y-axis vanishes when we include graph literacy as a variable, which confirms that people with a low graph literacy have more problems with interpreting misleading graphs [7, 10, 23].

All in all, our randomized experiment on a representative sample of the US population confirms the hypothesis by [18] that people are unfazed by misleading charts—except for people with a low graph literacy when it comes to making estimations.

One limitation of our study is that some of the participants did not seem to be used to working with percentages and gave unreasonably high estimates (for instance, one participant estimated 3,000% for an increase that was, in reality, 19.4%). An explanation for these very high estimates could be that the upper bound of the line graphs was 3,000. Maybe these participants thought the exercise was to state to which value the number of 'bluebeaks' had increased. These outliers did not influence our conclusions, and we kept them in our data set, but it might be good to realise that there might be a deeper misunderstanding hidden between the variables we measure.

Another limitation is that in our study large groups of participants had the same level of graph literacy. Even though the Short Graph Literacy test graphs are very basic graphs, 75%

percent of the participants answered only 1 or 2 questions correctly. This either means that many people do not know how to read and interpret simple graphs, which is in line with [24]. Furthermore it is in line with [25] who found statistical literacy in general, to be low, especially in the US population. It could also mean that the wording or topics of the exercises make them more difficult than they are supposed to be. Only a quarter of the participants demonstrated a decent graph literacy level. The Short Graph Literacy test [13] might be too short to separate between different levels of graph literacy. This might explain why graph literacy did not influence the judgement in our study. For follow-up studies, we would recommend using a more extensive graph literacy test.

For future research on this subject, it would also be interesting to see which kinds of contextualization lead to under or overestimating the correct answer. Furthermore, it might be good to see if these results hold for other graph types, such as bar charts.

Finally, we want to emphasize that although truncating the y-axis seems to have less influence on people's judgement and understanding than previously thought, it is in no way an excuse to make or use misleading graphs.

## Supporting information

**S1 File.**
(ZIP)

**S1 Appendix.**
(DOCX)

## Author Contributions

**Conceptualization:** Jannetje E. P. Driessen, Ionica Smeets, Casper J. Albers.

**Data curation:** Casper J. Albers.

**Formal analysis:** Daniël A. C. Vos, Casper J. Albers.

**Investigation:** Jannetje E. P. Driessen.

**Methodology:** Jannetje E. P. Driessen, Casper J. Albers.

**Software:** Jannetje E. P. Driessen, Daniël A. C. Vos.

**Supervision:** Ionica Smeets, Casper J. Albers.

**Writing – original draft:** Jannetje E. P. Driessen, Ionica Smeets.

**Writing – review & editing:** Daniël A. C. Vos, Ionica Smeets, Casper J. Albers.

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
