## [Decision Letter · Decision Letter 0]

9 Mar 2022

Misleading graphs in context: less misleading than expected

PONE-D-21-36742

Dear Dr. Albers,

We’re pleased to inform you that your manuscript has been judged scientifically suitable for publication and will be formally accepted for publication once it meets all outstanding technical requirements.

Kind regards,

Carlos Gracia-Lázaro

Academic Editor

PLOS ONE

Additional Editor Comments (optional):

This manuscript deals with an uncommon topic, and it has been hard finding reviewers.

Nevertheless, as editor, I have read the manuscript and agree with the unique reviewer's report.

Reviewers' comments:

Reviewer's Responses to Questions

**Comments to the Author**

1. Is the manuscript technically sound, and do the data support the conclusions?

Reviewer #1: Yes

2. Has the statistical analysis been performed appropriately and rigorously? 

Reviewer #1: Yes

3. Have the authors made all data underlying the findings in their manuscript fully available?

Reviewer #1: Yes

4. Is the manuscript presented in an intelligible fashion and written in standard English?

Reviewer #1: No

5. Review Comments to the Author

Reviewer #1: This is a straightforward study on an important practical issue in science communication. The authors use a representative sample of participants from the prodigy platform to examine the effect of what they call "context" (which in other fields in the social sciences is usually called "framing") and y-axis graph manipulation of people's quantitative estimates and goodness/badness judgments of a hypothetical scientific finding. The authors find that framing matters (people tend to judge the increase of bluebeaks as good when it is framed as an endangered species and bad when it is framed as a predator) but that neither graph literacy nor graph manipulation have an effect on qualitative judgment or relative accuracy of quantitative estimates.

Overall, the paper is well done, and it is based on reasonable hypotheses and expectations. The null finding regarding graph literacy and y-axis manipulation is a bit surprising. I commend the authors for making their data and code available which allows curious reviewers like myself to poke around. One thing I noticed when running exploratory models separate by gender is that there is a possible three-way interaction between y-axis manipulation, gender, graph literacy and judgment, such that graph literacy has a negative effect on the judgments of women in the shifted condition, but has a positive effect on men. Of course, this is a post hoc finding and the sample is small, but it could be something worthwhile to study in a more principled way in future work.

6. PLOS authors have the option to publish the peer review history of their article (what does this mean?). If published, this will include your full peer review and any attached files.

Reviewer #1: **Yes: **Omar Lizardo

---

## [Editor Report · Acceptance letter]

23 May 2022

PONE-D-21-36742 

Misleading graphs in context: less misleading than expected 

Dear Dr. Albers:

I'm pleased to inform you that your manuscript has been deemed suitable for publication in PLOS ONE. Congratulations! Your manuscript is now with our production department. 

Kind regards, 

on behalf of

Dr. Carlos Gracia-Lázaro 

Academic Editor

PLOS ONE